# Comprehensive Transcriptome Analysis in the Testis of the Silkworm, *Bombyx mori*

**DOI:** 10.3390/insects14080684

**Published:** 2023-08-02

**Authors:** Kohei Kakino, Hiroaki Mon, Takeru Ebihara, Masato Hino, Akitsu Masuda, Jae Man Lee, Takahiro Kusakabe

**Affiliations:** 1Laboratory of Insect Genome Science, Kyushu University Graduate School of Bioresource and Bioenvironmental Sciences, Motooka 744, Nishi-ku, Fukuoka 819-0395, Japan; k.kakino@agr.kyushu-u.ac.jp (K.K.); mhiro@agr.kyushu-u.ac.jp (H.M.); ebihara0915@agr.kyushu-u.ac.jp (T.E.); 2Laboratory of Sanitary Entomology, Kyushu University Graduate School of Bioresource and Bioenvironmental Sciences, Motooka 744, Nishi-ku, Fukuoka 819-0395, Japan; m.hino.018@agr.kyushu-u.ac.jp; 3Laboratory of Creative Science for Insect Industries, Kyushu University Graduate School of Bioresource and Bioenvironmental Sciences, Motooka 744, Nishi-ku, Fukuoka 819-0395, Japan; a.masuda@agr.kyushu-u.ac.jp (A.M.); jaemanle@agr.kyushu-u.ac.jp (J.M.L.)

**Keywords:** *Bombyx mori*, transcriptome, testis

## Abstract

**Simple Summary:**

Spermatogenesis, an important process in reproduction, is conserved across species but has unique aspects in silkworms, such as the maintenance of the spermatogonium by a somatic single cell and coordinated fertilization by nucleated and anucleated sperm. Despite some histological research, the molecular mechanisms of silkworm spermatogenesis remain largely unexplored. Taking advantage of advances in RNA-seq analysis technology, this study characterizes genes expressed in silkworm testis via a comprehensive transcriptome analysis, contrasting different tissues and regions within the testis and making comparisons with fruit flies. Our investigation revealed extensive gene expression (86.5% of all genes) in the silkworm testis, the highest number of tissue-specific genes among examined tissues in silkworm testis. We also identified region-specific enriched genes in undifferentiated and differentiated germ cells. Moreover, cross-species transcriptome comparisons confirmed conserved gene expression patterns. Further analysis of region-specific enriched genes in the silkworm testis revealed the enrichment of genes associated with the biological process, corroborating findings in silkworms and other organisms. This study extends our understanding of silkworm spermatogenesis and fills an important gap for future investigations.

**Abstract:**

Spermatogenesis is an important process in reproduction and is conserved across species, but in *Bombyx mori*, it shows peculiarities, such as the maintenance of spermatogonia by apical cells and fertilization by dimorphic spermatozoa. In this study, we attempted to characterize the genes expressed in the testis of *B. mori,* focusing on aspects of expression patterns and gene function by transcriptome comparisons between different tissues, internal testis regions, and *Drosophila melanogaster*. The transcriptome analysis of 12 tissues of *B. mori*, including those of testis, revealed the widespread gene expression of 20,962 genes and 1705 testis-specific genes. A comparative analysis of the stem region (SR) and differentiated regions (DR) of the testis revealed 4554 and 3980 specific-enriched genes, respectively. In addition, comparisons with *D. melanogaster* testis transcriptome revealed homologs of 1204 SR and 389 DR specific-enriched genes that were similarly expressed in equivalent regions of *Drosophila* testis. Moreover, gene ontology (GO) enrichment analysis was performed for SR-specific enriched genes and DR-specific enriched genes, and the GO terms of several biological processes were enriched, confirming previous findings. This study advances our understanding of spermatogenesis in *B. mori* and provides an important basis for future research, filling a knowledge gap between fly and mammalian studies.

## 1. Introduction

Gametogenesis is a critical process for the survival of any species and is carried out differently in different organisms while maintaining its essential reproductive function. Spermatogenesis is an important biological process for developing mature spermatozoa from germ cells. Well-orchestrated molecular and cellular events characterize this complex process, including mitosis, meiosis, and cellular morphogenesis [1,2].

The testis of the silkworm, *Bombyx mori*, consists of four follicles and several cell types that contribute to spermatogenesis. The stem cell spermatogonium, formed from primordial germ cells, undergoes mitosis to produce 64 spermatocytes. These spermatocytes undergo two meiotic divisions to produce 256 spermatozoa. At the tip of each follicle, a somatic single cell is known as an apical cell [3,4]. The apical cell is considered to function as a niche cell for stem-like spermatogonium [4]. The proliferating germ cells around the apical cell migrate to the basal region of the testis via somatic cell division, meiosis, and sperm morphogenesis. The spermatids undergo morphogenesis and elongation, eventually developing into mature sperm. *B. mori* spermatogenesis is characterized by the production of two types of sperm bundles, namely nucleate eupyrene sperm and anucleate apyrene sperm, which diverge according to the timing of elongation [3,4,5,6,7].

Genomic studies have identified several genes that are expressed in a testis-specific manner in *B. mori*, such as β-tubulin, BmDmc1, and BmAHA1. Although these genes show specific expression patterns in the testis, their direct involvement in the process of spermatogenesis has not been definitively established [8,9,10]. In recent years, the advancement of genome editing technologies like TALEN and CRISPR/Cas9 has significantly simplified the process of analyzing *B. mori* in vivo. Along with this progress, it has been reported that the *B. mori* genes BmPMFBP1, BmSxl, and BmPnldc1 play a direct role in the process of spermatogenesis. Specifically, these genes play an important role in dimorphic spermatogenesis observed in lepidopteran insects [11,12,13]. Despite these advancements, the investigation of spermatogenesis-related genes in *B. mori* remains a challenging endeavor due to the complexity of the process and the intricate nature of conducting gene-specific studies.

Over the past several decades, significant progress has been made in understanding the molecular mechanisms of spermatogenesis, the complex physiological process that produces mature sperm. The fruit fly, *Drosophila melanogaster*, has emerged as a powerful model organism for studying this process due to the availability of advanced genetic tools and the high degree of conservation of spermatogenesis-related genes between flies and mammals [14,15]. In recent years, comprehensive transcriptomic analyses using RNA-seq, including single-cell RNA-seq, have significantly advanced our knowledge of spermatogenesis. These techniques have allowed us to better understand the complex regulation of gene expression at various stages of spermatogenesis [16,17,18]. Thus, while spermatogenesis in *D. melanogaster* has been extensively studied, spermatogenesis in other insects, such as *B. mori*, is not comprehensively understood. The *B. mori* genome was first decoded in 2004 [19,20], and several researchers subsequently performed transcriptome analyses to better understand its gene expression and its regulatory mechanisms [21,22]. However, the details of the molecular mechanism of spermatogenesis in *B. mori* remain unresolved, although the mechanism identified in *D. melanogaster* has been plausibly applied because the spermatogenesis mechanism is conserved in many organisms. Therefore, we believe it is important to compare gene expression profiling data from the testes of two model insects, *B. mori* and *D. melanogaster*, due to their genetic and biological similarities yet distinct differences. This approach would provide a robust tool for elucidating the mechanisms of spermatogenesis in *B. mori* and identify genes with similar or conserved expression patterns among these species. In particular, genes that have already been reported to be involved in spermatogenesis in *D. melanogaster* and have similar expression patterns in *B. mori* and *D. melanogaster* testes are likely to have maintained their function throughout evolution, suggesting that they will play a similar role in *B. mori*.

In this study, we extensively profiled gene expression in *B. mori* testes to broaden our understanding of expressed genes. We compared gene expressions across different tissues and germ cell differentiation stages within the *B. mori* and with data from *D. melanogaster*. In these analyses, we strived to clarify which genes expressed in the testes are testis-specific, which are involved in particular stages of spermatogenesis, and which have their functions already elucidated in *D. melanogaster*. Our insights will provide foundational information, enhancing the understanding of the molecular mechanisms of spermatogenesis in *B. mori*.

## 2. Materials and Methods

### 2.1. RNA-seq Data of B. mori Tissues

RNA-seq raw data of silkworm, *B. mori*, fifth instar larvae day 3 samples (12 tissues, 35 samples) for comparison of gene expression between tissues were downloaded from NCBI (accession number: PRJNA559726). Accession numbers for the data used are listed in Appendix A.

### 2.2. RNA Extraction and RNA Sequencing

The *B. mori* p50T strain was obtained from the Institute of Genetic Resources, Kyushu University Graduate School. The larvae were reared on mulberry leaves at 25 °C. RNA-seq testing was performed using *B. mori* testes from day 3 of 5th instar larvae. First, testes removed from larvae were dissected in PBS, collected under microscope, and separated into germinal proliferation centers (called stem region) and other content cells (differentiated region), respectively, in 3 replicates. Total RNA was extracted from each sample using ISOGEN reagent (Nippon Gene, Toyama, Japan) and purified using RNeasy ^®^ Plus universal mini kit (Qiagen, Hilden, Germany) according to the manufacturer’s instructions. The RNA content and purity of the extracted RNA were analyzed using a NanoDrop One spectrophotometer (Thermo Fisher Scientific, Waltham, MA, USA) and further quantified using a Quantus Fluorometer (Promega, Madison, WI, USA) according to the manufacturer’s instructions. The extracted RNA was then sequenced using the low-cost and simple RNA-Seq (Lasy-Seq) method [23]. The resulting sequence data have been deposited at DDBJ/EMBL/GenBank under the accession DRR492946-DRR492951.

### 2.3. Morphological Observation of B. mori Testis

Testes were carefully dissected from fifth instar, day 3 *B. mori* and then fixed using 4% paraformaldehyde in 1 × PBS. Following this, we prepared paraffin sections from the collected testes and performed Hematoxylin and Eosin (HE) staining. Subsequent observations were conducted using a Leica M205A stereomicroscope (Leica).

### 2.4. RNA Sequencing (RNA-seq) Experiment

The raw RNA-seq data were trimmed, and the quality was confirmed using fastp (v-0.12) [24]. The clean reads were directly aligned and quantified using Salmon (v-1.8.0) [25] to the silkworm reference transcripts data, which were downloaded from NBDC (https://dbarchive.biosciencedbc.jp/en/kaiko/desc.html (accessed on 3 February 2022)). The expression data of transcript level, which contained several transcriptional variants, conversed to gene level using Tximport (v-1.22.0) [26]. Differential expression genes (DEGs) of each sample were determined by DESeq2 (v-1.34.0) [27]. Coding regions within the transcripts were predicted and extracted using the TransDecoder (v-5.5.0: https://github.com/TransDecoder/TransDecoder/wiki/ (accessed on 21 April 2023)) tool with the default parameters.

### 2.5. Statistical Calculations for Tissue-Specific and Region-Enriched Gene Calculations

Tissue-specific gene calculation: Pairwise comparisons between different tissues were performed using the Wald test via the DESeq2 package. Genes were considered significant if they had an adjusted *p*-value (*p*-adj) of less than 0.05 and a log2FoldChange of less than 0.5. Genes that met these criteria in all tissue comparisons were classified as tissue-specific. Calculation of region-enriched genes: A procedure similar to step 1 was followed, but with a focus on comparisons between the stem region (SR) and the differentiated region (DR). Genes that showed significance (*p*-adj < 0.05, log2FoldChange < 0.5) in the SR vs. DR comparison were considered SR-enriched or DR-enriched, respectively. The above procedures were used to identify genes specifically expressed or enriched in different tissues and regions.

### 2.6. Comparative Analysis of Total Genes and Transcripts Between D. melanogaster and B. mori 

Comparative analysis was performed between the total genes of fruit fly, *D. melanogaster* and the total transcripts of silkworm, *B. mori*. *D. melanogaster* total genes were downloaded from flybase (http://ftp.flybase.net/releases/FB2022_03/dmel_r6.46/ (accessed on 1 April 2022)). The tblastx function of the BLAST tool was used for this comparison, with an e-value cut-off threshold of 1 × 10^−5^.

For each transcript, the *D. melanogaster* gene with the lowest e-value was annotated as its homolog. In cases where multiple transcripts were assigned with a single *B. mori* gene and annotated with different *D. melanogaster* gene homologs, the homolog with the lowest e-value was selected as the representative homolog for that gene.

### 2.7. GO Enrichment Analysis

The GO term data for the *B. mori* genes were downloaded from SGID (http://sgid.popgenetics.net/ (accessed on 8 May 2022)) and adapted to this study’s gene ID [28]. The GO enrichment analysis was implemented using Goatools, a Python library for gene ontology, under default conditions. The Benjamini–Hochberg (BH) method was employed to adjust the *p*-values, which helps control the False Discovery Rate [29]. 

## 3. Results and Discussion

### 3.1. Comparative Analysis of Gene Expression among B. mori Tissues

First, we attempted to characterize genes expressed in the testis by comparing gene expression among other *B. mori* tissues. As described under the Materials and Methods, raw RNA-seq data were obtained from 12 different *B. mori* tissues (anterior silk glands, epidermis, fat body, head, hemolymph, malpighian tubule, middle silk glands, midgut, ovary, posterior silk glands, testis, and trachea) from the NCBI database. The raw data were then processed using the fastp program for trimming and the Salmon program for mapping and quantification against *B. mori* reference transcripts. The R package Tximport program was then used to calculate gene expression levels in each tissue. For genes with multiple transcript variants, they were combined to represent a single gene, and the expression level for each gene was calculated. As a result, according to this definition, 51,926 total transcripts were integrated for *B. mori*, resulting in 24,229 total genes (Appendix A). The obtained gene expression data for each tissue were subjected to comparative analysis using DESeq2.

To understand the differences in gene expression between tissues, we performed PCA analysis and found that the expression levels of genes in “testis” and “midgut” were different from the other ten tissues on the PC1 and PC2 axes (Figure 1A). Then, the number of genes expressed in each sample were counted, considering any gene with a count of one or more as expressed. The number of genes expressed in each tissue was counted as an average across samples for each gene (count ≧ 1). The results showed that the average number of genes expressed among the tissues was 16,733. Comparing the number of genes expressed in each tissue, the posterior silk gland expressed the least number of genes, 14,354. On the other hand, the largest number of genes expressed was in “testis”, with 20,962 genes expressed, representing 86.5% of the total genes. (Figure 1B, Table 1). Next, we attempted to calculate tissue-specific expressed genes. This was performed by comparing the expression of genes in a specific tissue to their expression in all other tissues. We defined tissue-specific genes as those that had a *p*-value of less than 0.05 and a log2FoldChange of less than 0.05 across all tissues. Using this definition, we identified 1705 genes as being specifically expressed in the testis (Figure 1C, Table 1 and Appendix A). 

Detailed analysis of testis RNA-seq data revealed that 20,962 genes, comprising 86.5% of all genes, were expressed in the testis. This finding aligns with reports in flies and mammals, showing a large number of gene species transcribed in the testis [16,30]. The extensive gene expression in the testis corresponds to a range of biological processes. For instance, during spermatogenesis, germ cells undergo extensive chromatin remodeling, leading to an overall permissive chromatin state and widespread transcriptional facilitation. This process is not limited to coding proteins but involves long non-coding RNAs [16,30]. In the present study, we sorted the expressed genes into coding and non-coding categories. Of the 20,962 genes confirmed to be expressed in the testis, 13,111 were protein-coding genes, 6423 were non-coding genes, and an intriguing set of 1428 genes appeared to potentially possess both coding and non-coding variants (Appendix A).

Simultaneously, new genes emerge via various mechanisms like gene shuffling, gene fission/fusion, retrotransposition, duplication–divergence, and lateral gene transfer. Recent studies have highlighted the de novo creation of genes with open reading frames from non-coding genes [31,32,33,34,35]. Expression patterns of such young genes are often more tissue- or condition-specific compared to established genes. Particularly, the high expression of de novo genes has been observed in male reproductive tissues across multiple species, including flies, mice, and humans [16,34,35,36,37,38]. Our study revealed a substantial number of 1705 genes with tissue-specific expression in the testis. Mon et al. (2022) identified 272 lepidopteran-specific uncharacterized with no Pfam domain by in silico analysis [39]. Compared to the testis-specific expression of *B. mori* in this study, 47 of the 272 genes were expressed in a testis-specific manner (Appendix A). Of the 1705 testis-specific genes, 270 genes were found that all the gene variants were non-coding transcripts, and the other 208 genes contained non-coding transcripts in some of the gene variants. Xia et al. (2020) suggest that the widespread transcription during spermatogenesis maintains DNA sequence integrity in the male germline by correcting DNA damage by a mechanism we term “transcriptional scanning”. They also propose that transcriptional scanning regulates mutation rates in a gene-specific manner, preserving DNA sequence integrity for most genes while accelerating evolution for a specific subset [30]. The comprehensive examination of *B. mori* testis gene expression in this study allowed us to detect phenomena similar to those reported in flies and mammals (widespread gene expression and expression of many testis-specific genes, including lineage-specific uncharacterized genes and non-coding genes), confirming that the principles are conserved in the *B. mori*.

### 3.2. Transcriptome Analysis of Silkworm, B. mori, and Testis between Different Regions

Spermatogenesis occurs via a series of processes that begin with stem-cell-like spermatogonium cells, undergo multiple somatic cell divisions and meiosis to become spermatocytes, and then undergo morphological changes to become spermatozoa. A comparison between tissues allowed us to detect the characteristics of genes expressed in the testis and genes specific to the testis, but this information was not sufficient to characterize the stages of spermatogenesis. Therefore, to obtain more detailed information on expressed genes during spermatogenesis in the *B. mori* testis, the testis was divided into the stem region (SR) and the differentiated region (DR) and sampled for RNA-seq analysis (Figure 2A). As described in the Introduction, germ cells are surrounded by somatic cells called cyst cells, which form a seminal vesicle in the *B. mori* testis. Within this seminal vesicle, germ cells undergo meiosis, somatic cell division, and sperm elongation. The SR contains spermatogonia stage spermatocysts and apical cells (niche cells). The DR, on the other hand, is enriched with spermatocysts containing mitotic spermatocytes, meiotic spermatocytes, and elongating spermatocytes.

To compare the gene expression profiles between the SR and DR regions, which are understood to be at distinct stages of morphological differentiation, we performed RNA-seq analysis. The RNA-seq analysis was performed using mRNAs extracted from SR and DR in biological triplicates. The 3’-RNA-seq method was performed using the Lasy-seq method and HiSeq 2500, a high-throughput library preparation. We processed the obtained raw data using fastp, Salmon, and Tximport to transform it into gene-level expression data. We then carried out a comparative analysis of gene expression across different regions utilizing DESeq2. The results confirmed that 17,983 genes were expressed in the SR and that 16,717 genes were expressed in the DR in one or more read counts. And 18,803 genes were identified as being expressed in at least one of the different regions of the testis (Figure 2B, Appendix A). Of these, 18,034 genes were also found to be expressed in previous RNA-seq studies, as shown by the testis transcriptome data used for comparison between tissues, and the other 769 genes were discovered to be expressed in the present RNA-seq study (Appendix A). In the SR and DR, 2086 and 820 genes were specifically expressed, respectively (Figure 2B). The pattern of the transcriptional accumulation of each gene was then determined, and genes showing significant differences in transcription levels between regions were filtered based on DESeq2. As a result, we identified 4554 and 3980 genes with specific enriched expression in the SR and DR, respectively (Figure 2C, Appendix A). Furthermore, among these region-specific enriched genes, we found that 89 in the SR and 1327 in the DR were specifically expressed in the testis (Figure 3).

### 3.3. Comparative Analysis of Gene Expression in Histologically and Functionally Homologous Regions of B. mori and D. melanogaster Testis

Next, we examined whether the expression profiles of these genes, which were significantly highly expressed in each region of the *B. mori* testis, were conserved in *D. melanogaster*, the same insect order and a model organism for the studies of spermatogenesis. V. Vedelek et al. (2018) performed transcriptome analysis of the *D. melanogaster* testis, dividing it into the “apical region”, which contains stem cells and developing spermatocytes; the “middle region”, with an enrichment of meiotic cysts; and the “basal region”, which contains elongated post-meiotic cysts with spermatids [18]. Considering the stages of germ cell differentiation enriched in each region of both species, the *B. mori* “stem region (SR)” corresponds to the “apical region” of *D. melanogaster*, and the *B. mori* “differentiated region (DR)” corresponds to the “middle and basal regions” of *D. melanogaster*. This comparison prompted us to compare the expressed genes between these homologous regions to investigate whether homologous genes are also expressed between them.

A BLAST search was performed to investigate homology between all *B. mori* genes and *D. melanogaster* genes. As a result, it was found that 14,447 genes, or 59.6% of all *B. mori* genes, had homology with *D. melanogaster* genes (Appendix A). It is important to note that the term “homologous” in this context could include not only orthologs but also paralogs. Of the 4554 SR-specific enriched *B. mori* genes, 3151 were homologous to 2640 *D. melanogaster* genes. A comparison of these 3151 genes with the 3214 genes significantly highly expressed in the *D. melanogaster* “apical region” revealed that the 1024 *D. melanogaster* homologues of the 1204 *B. mori* genes are also highly expressed in the homologous region of the *D. melanogaster* testis (Figure 4A, Appendix A). We next compared the gene expression in the “differentiated region” (DR) of *B. mori* with the corresponding “middle and basal regions” in *D. melanogaster*. Of the 3980 genes specifically enriched in the *B. mori* DR, 2127 were homologous to 1757 *D. melanogaster* genes. When these 2127 genes were compared with the 2152 genes significantly highly expressed in the *D. melanogaster* “middle and basal regions”, it was revealed that the 314 *D. melanogaster* homologs of the 389 *B. mori* genes are also highly expressed in the homologous region of the *D. melanogaster* testis (Figure 4B, Appendix A).

### 3.4. Gene Ontology (GO) Enrichment Analysis of Differentially Expressed Genes (DEGs) in B. mori Testis

Comprehensive differentially expressed genes (DEGs) analyses were performed from multiple perspectives to profile *B. mori* testes by gene expression. Gene ontology (GO) enrichment analysis was performed to define the DEGs from these analyses further, focusing specifically on DEGs that show notable variation between stem and differentiated regions within the *B. mori* testis. The GO terms for the *B. mori* gene sets used in this study were annotated using the GO term sets for the genes assigned to the SGID. Subsequently, GO enrichment analysis was performed using Goatools (*p* < 0.05).

#### 3.4.1. GO Enrichment Analysis of SR-Specific Enriched Genes

In the 4554 SR-specific enriched genes, there were 807 GO terms related to biological process enrichment. In the 807 GO terms, focusing on higher-level terms, there were significant enrichments in “metabolic process”, “cellular process”, “biological regulation”, “developmental process”, “reproductive process”, “growth”, and “homeostatic processes” (Figure 5A, Appendix A). 

In our study, “RNA metabolic process” (GO:0016070) emerged as a notable category among “metabolic process” and “cellular process” with significant enrichment in the SR (*p* = 4 × 10^−78^). It encompassed a total of 1025 genes out of the 4554 genes specifically enriched in the SR (Appendix A). These 1025 genes were subjected to enrichment analysis, followed by a cross-reference check to determine overlap with 1204 genes that showed increased expression in the SR apical regions of both *B. mori* and *D. melanogaster*. This led to the identification of 488 genes related to “RNA metabolic process” that showed a similar expression landscape in the testes of both species (Appendix A). Importantly, these 488 genes included homologs of Grip75 [40] and maleless [41], all established players in *D. melanogaster* spermatogenesis. 

Within the SR, the term “regulation of RNA metabolism” (GO:0051252), a sub-category of “biological regulation”, emerged as another significantly enriched GO term (*p* = 1.6 × 10^−17^). It correlated with 456 genes from the pool of 4554 SR-specific enriched genes (Appendix A). These 456 genes were subjected to a focused analysis, followed by another round of cross-referencing to determine overlap with the 1204 genes with increased expression in the SR apical regions of *B. mori* and *D. melanogaster*. This led to the discovery of 226 genes associated with “regulation of RNA metabolism” that showed similar expression patterns in the testes of both species (Appendix A). Notably, these 226 genes included homologs of Brahma-associated protein 55kD [42] and Zn finger homeodomain 1 [43], all of which have been shown to directly influence spermatogenesis in *D. melanogaster*.

Our study highlights a significant enrichment of genes associated with “RNA metabolic processes” or “regulation of RNA metabolic processes” in the SR of *B. mori* testes. Previous investigations in *Drosophila* and mammals have emphasized the crucial role these genes play in spermatogenesis [14,15,44,45,46,47,48,49,50]. The involvement of RNA metabolic processes is pivotal for regulating gene expression, which is integral for the functionality of germline stem cells (GSCs) and their maturation into spermatozoa [48,49]. In line with this, RNA-binding proteins have been established as key determinants of small RNA and mRNA profiles in *Drosophila* testes [50]. Reflecting on these observations, we suggest that *B. mori* genes implicated in “RNA metabolic processes” or “regulation of RNA metabolic processes” might be crucial for germline stem cells’ proper growth, differentiation, and upkeep, thereby contributing to normal spermatogenesis. 

Turning our focus to the “meiotic cell cycle process” (GO:0051321), this term, a category under “metabolic process” and “cellular process”, also revealed significant enrichment (*p* = 2.8 × 10^−8^), relating to 54 out of the 4554 SR-specific enriched genes (Appendix A). To delve deeper, we sought out any shared genes between these 54 and the 1204 genes, which exhibited similar heightened expression in the SRs of both *B. mori* and *D. melanogaster*. This yielded 35 genes linked to the “meiotic cell cycle process” that showed analogous expression patterns across the testes of both species (Appendix A). Notably, these included homologs of Rab11 [51] with validated roles in *D. melanogaster* spermatogenesis.

Another “cell cycle process” term, “mitotic cell cycle process” (GO:1903047), also emerged as significantly enriched (*p* = 1.2 × 10^−11^) and was linked to 91 out of the 4554 SR-specific enriched genes (Appendix A). Similarly, a hunt for shared genes between these 91 and the 1204 genes, which showed similar heightened expression in *B. mori* and *D. melanogaster* SR, led to the identification of 54 genes. These genes, related to the “mitotic cell cycle process”, exhibited analogous expression patterns in the testes of both species (Appendix A). Intriguingly, this list includes homologs of Bub1 kinase [52] and tumbleweed [53], each with well-established roles in *D. melanogaster* spermatogenesis.

Comparable to the *D. melanogaster* testis apex, the *B. mori* testis SR also harbors proliferative spermatogonia and somatic cyst cells around an apical cell, analogous to the hub cells in *D. melanogaster*. Every spermatogonium, upon encapsulation by cyst cells, multiplies by mitosis into 64 cells before proceeding to meiosis [4,54,55]. The observation of genes associated with “meiotic cell cycle process” and “mitotic cell cycle process” in our study aligns well with previous histological studies in *D. melanogaster* and *B. mori* testes, reinforcing their critical role in gene expression during spermatogenesis.

#### 3.4.2. GO Enrichment Analysis of DR-Specific Enriched Genes

In contrast, 3980 genes enriched in the DR of the *B. mori* testis showed enrichment in 87 biological process GO terms, with higher-levels terms of “microtubule-based process”, “cellular component organization or biogenesis”, “cell motility”, “cellular localization”, and “regulation of molecular functions and protein-containing complex localization” (Figure 5B, Appendix A).

In our investigation of DR-specific enriched genes, “cilium assembly” (GO:0060271) emerged as the topmost enriched term (Appendix A). This subcategory of “cellular component organization or biogenesis” includes 66 of the total 3980 DR-specific enriched genes (*p* = 1.6 × 10^−18^). Subsequently, a comparative search was conducted between these 66 genes and 389 genes that demonstrate a similar expression enrichment in both the *B. mori* DR and *D. melanogaster* middle and basal regions. This comparison led to the discovery of 12 genes linked with the “Cilium assembly” term, manifesting equivalent expression patterns in the testes of both *B. mori* and *D. melanogaster* (Appendix A). Notably, this subset of 12 genes featured kinesin-like protein at 10A [56], male fertility factor kl2 [57], and male fertility factor kl3 [57], all of which bear direct implications in *D. melanogaster* spermatogenesis. In addition, exploring the tissue specificity of the 66 genes incorporated in “cilium assembly”, we ascertained that 46 of them were testis-specific in their expression (Appendix A). 

The differentiation region (DR) of *B. mori* testes is composed of cells progressing into advanced stages of spermatogenesis, including spermatocytes, spermatids, and elongated spermatozoa (Figure 2A). With the indispensable role of cilia and flagella in successful sperm fertilization, the criticality of cilium assembly in spermatogenesis is evident [2,58,59]. The substantial gene pool in the DR related to ciliary assembly affirms this region’s pivotal role in spermatogenesis progression. Additionally, we found spermatogenesis to be a unique process within the testis, typically characterized by a testis-specific expression of associated genes.

Turning our focus to “microtubule cytoskeleton organization” (GO:0000226), a subterm of both “microtubule-based process” and “cellular component organization or biogenesis”, we identified significant enrichment within the DR (*p* = 3.9 × 10^−8^) (Appendix A). This term incorporated 80 of the 3980 DR-specific enriched genes. Further tissue-specific examination of these 80 genes revealed that 46 genes exhibited testis-specific expression. A subsequent comparison between these 80 genes and the 389 genes with similar enriched expression in both the *B. mori* DR and *D. melanogaster* middle and basal regions unveiled 24 genes associated with the RNA metabolic process term, echoing analogous expression patterns in the testes of both species (Appendix A). Dynein light chain 90F [60], Kinesin-like protein at 3A [61], and Lkb1 kinase [62] were among these 24 genes, each playing a crucial role in *D. melanogaster* spermatogenesis. 

Microtubules, integral constituents of the cytoskeleton, comprise protein filaments that offer structural support, maintain cellular shape, and expedite various cellular processes like cell division, intracellular transport, and cell motility. During spermiogenesis, the final leg of spermatogenesis, spermatids transform by a sequence of morphological modifications to become elongated spermatozoa [63,64]. Moreover, in *B. mori*, microtubules contribute to the assembly of the contractile ring in spermatocytes. The distribution of actin in live *B. mori* spermatocytes is manipulated by varying configurations of microtubules, a critical factor for cytokinesis [65,66]. Given this, in the DR, it stands to reason that genes related to microtubule cytoskeleton organization find expression and likely play a significant role in comprehensive spermatogenesis.

## 4. Conclusions

In this study, we performed a comprehensive transcriptome analysis to characterize the genes expressed in *B. mori* testes. In particular, RNA-seq analysis performed by dividing testes into SR and DR according to the germ cell differentiation stage was a novel approach used for *B. mori* testes and provided detailed transcriptome profiles. Based on this profile, we performed a comparative analysis with the *D. melanogaster* testis transcriptome, which allowed us to identify genes specifically enriched in homologous regions across species. In addition, the genes specifically enriched in the SR and DR were confirmed to be related to the biological processes essential for each region. This research provides comprehensive transcript information for the *B. mori* testis and bridges the knowledge gap between studies on *Drosophila* and mammals by comparing it with advanced spermatogenesis research in these organisms, thus providing a crucial foundation for future studies on *B. mori* spermatogenesis.

## Figures and Tables

**Figure 1 insects-14-00684-f001:**
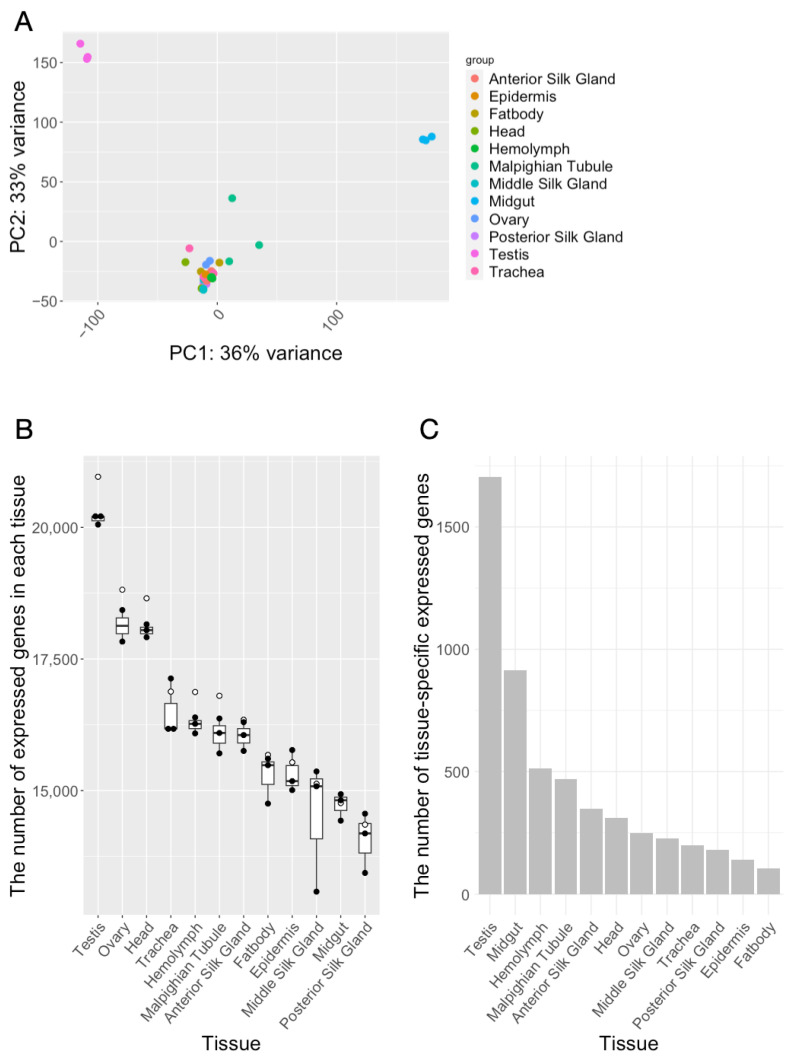
Comparative analysis of gene expression among *B. mori* tissues using RNA-seq. (**A**) Principal component analysis (PCA) of the transcriptome data. Each dot in the plot represents a distinct sample, with 12 tissues of the *B. mori* represented: anterior silk glands, epidermis, fat body, head, hemolymph, malpighian tubules, middle silk glands, midgut, ovary, posterior silk glands, testis, and trachea. (**B**) The number of expressed genes in each tissue. The number of expressed genes per sample is shown in the box-and-whisker diagram. White circles indicate the number of genes that had a mean value ≧ 1 between samples of the same tissue. (**C**) The number of tissue-specific expressed genes.

**Figure 2 insects-14-00684-f002:**
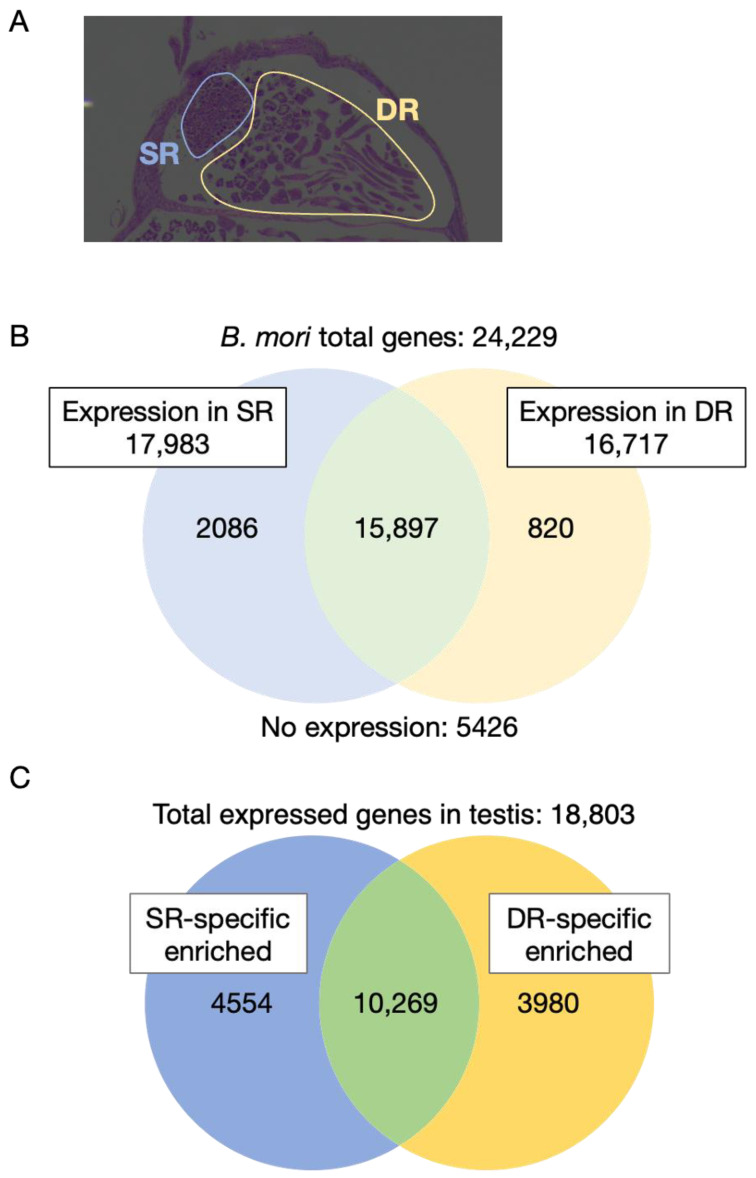
Transcriptome analysis of *B. mori* testis between different regions. (**A**) Structural diagram of *B. mori* testis (SR: stem region; DR: differentiated region). (**B**)Venn diagram of the number of genes expressed in the SR and DR (count ≧ 1). (**C**) Venn diagram of the number of SR- or DR-specific enriched expressed genes (*p* < 0.05).

**Figure 3 insects-14-00684-f003:**
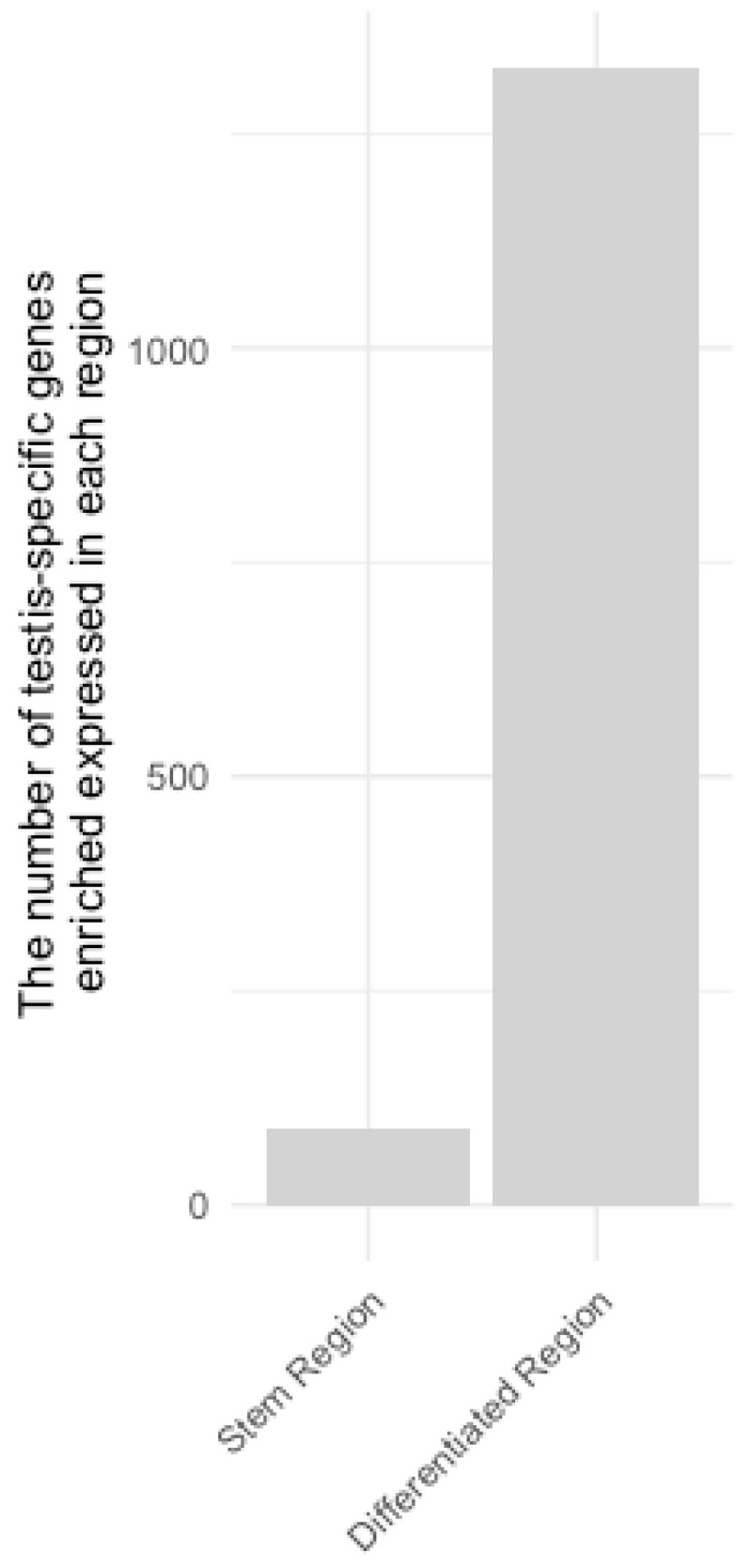
The number of testis-specific genes with SR- or DR-specific enriched expression.

**Figure 4 insects-14-00684-f004:**
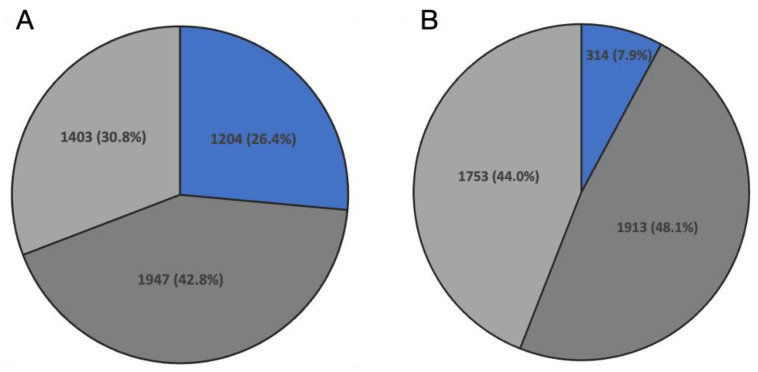
Comparative analysis of gene expression in histologically and functionally homologous regions of *B. mori* and *D. melanogaster* testes. (**A**) Stem region in *B. mori* testis vs. apical region in *D. melanogaster* testis. (**B**) Differentiated region in *B. mori* testis vs. middle and basal region in *D. melanogaster* testis. Blue: The number of genes with similar region-specific enriched expression pattern in both *B. mori* and *D. melanogaster*. Light grey: The number of genes with region-specific enriched expression pattern in only *B. mori*. Dark grey: The number of genes with region-specific enriched expression pattern in only *D. melanogaster*.

**Figure 5 insects-14-00684-f005:**
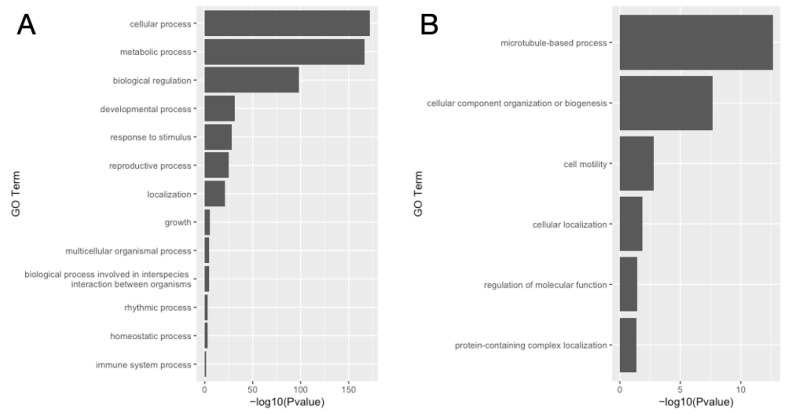
Gene ontology enrichment analysis of DEGs among SR and DR. (**A**) Stem region. (**B**) Differentiated region.

**Table 1 insects-14-00684-t001:** The number of expression genes under each classification.

Tissue	Expressed Genes	Tissue-Specific Expressed Genes
Anterior Silk Gland	16,346	349
Epidermis	15,537	139
Fat body	15,677	105
Head	18,653	310
Hemolymph	16,873	513
Malpighian Tubule	16,801	470
Middle Silk Gland	15,130	227
Midgut	14,763	915
Ovary	18,816	248
Posterior Silk Gland	14,354	179
Testis	20,962	1705
Trachea	16,881	198

## Data Availability

The raw RNA-seq data used in this study are available from accession number PRJDB16223.

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
