# Peer review of "Comprehensive Transcriptome Analysis in the Testis of the Silkworm, Bombyx mori"

_insects, 2023, doi:10.3390/insects14080684_

Round 1

Reviewer 1 Report

The manuscript entitled “Comprehensive transcriptome analysis in the testis of the silkworm, Bombyx mori” by Kakino et al. has compared the differential expression of different tissues of the silkworm by transcriptome analysis, which showed that lots of specific genes were expressed in testis. The study further deciphered the expression patterns between stem region and differentiated region of the testis, and found very distinct expression in different regions. Moreover, they compared the homologous of Bombyx mori and Drosophila melanogaster testis, and revealed their functional conservation throughout evolution. The study has given us some interesting clues to understand the molecular mechanisms of spermatogenesis in the silkworm. The manuscript is well-organized, and I have only several concerns as follows: 

(1) Figure 1, I suggested that the authors should remove the underline in the names of tissues. The orders of tissues in B and C could be arranged from high to low to show a nice figure.

(2) Figure 2, the photos of stem region and differentiated region and morphological observation of the testis should be shown in the main figure rather than in supplements. In addition, it was difficult to understand the “significantly differences” and “Highly expression” in B, the gene numbers should be edited in venn diagram, and the words were wrong.

(3) Figure 3, how about the downregulated genes in SR and DR?

(4) Figure 4, the authors should explain the relationship among the regions of SR and DR in silkworm with the regions of apical, middle, and basal in fruit fly, how to compare these different regions.

(5) The format and mistakes throughout the manuscript should be carefully checked.

NA

Reviewer 2 Report

Gene expression pattern of testise was already reported, so I didn't feel novelity in this paper. However, comparative analysis of stem region (SR) and differentiated regions (DR) were firstly reported in this paper, therefore, it may be novel in that respect.  Please clearly state the novelty of this paper in the conclusion.

Was there any difference in expression between  SR and DR for the gene mentioned in the introduction? 

There are many careless mistakes in this paper, so please check your paper before submit.

Please unify description "Fig" or "Figure"

Please check space before unit.

Please check  abbreviated scientific name, such as Bombyx mori or B. mori

Check the italicized description of the scientific name.

Line: 108, D. melanogaste 

Please unify the description of references.
